# Comparative Assessment of the Functional Parameters for Metal-Ceramic and All-Ceramic Teeth Restorations in Prosthetic Dentistry—A Literature Review

**DOI:** 10.3390/biology11040556

**Published:** 2022-04-05

**Authors:** Ana Ispas, Laura Iosif, Daniela Popa, Marius Negucioiu, Mariana Constantiniuc, Cecilia Bacali, Smaranda Buduru

**Affiliations:** 1Department of Prosthodontics, Faculty of Dental Medicine, “Iuliu Haţieganu” University of Medicine and Pharmacy, 400006 Cluj-Napoca, Romania; ispas.ana@umfcluj.ro (A.I.); popa.daniela@umfcluj.ro (D.P.); marius.negucioiu@umfcluj.ro (M.N.); mconstantiniuc@umfcluj.ro (M.C.); cecilia.bacali@umfcluj.ro (C.B.); dana.buduru@umfcluj.ro (S.B.); 2Department of Prosthodontics, Faculty of Dental Medicine, “Carol Davila” University of Medicine and Pharmacy, 010221 Bucharest, Romania

**Keywords:** all-ceramic, metal-ceramic, survival, biological complication, technical complication, fixed partial denture

## Abstract

**Simple Summary:**

In the last decades, the science and technology of all-ceramic teeth restorations witnessed the fastest-growing field of restorative materials in research and development for fixed prosthodontics. Due to their properties, characterized by a high level of biocompatibility, excellent optical properties, and high fracture resistance, all-ceramic material can also be used in the posterior areas of the dental arches. However, metal-ceramic fixed partial dentures are still perceived as the golden standard for those restorations, thanks to the positive clinical outcomes. Irrespective of the materials of which fixed partial dentures are performed, their success rates depend on the expected outcome and on how they perform in the oral environment. Such conditions of failure restorations may include biological factors (i.e., secondary decay, loss of vitality, periodontal disease, or abutment fracture) or technical factors (i.e., chipping, ceramic fracture, framework fracture, or loss of retention). Our aim is to provide the current evidence for the efficacy of metal-ceramics and all-ceramics in the treatment of multiple posterior edentulous spaces. Moreover, we aim to compare the survival rates of metal-ceramic and all-ceramic fixed partial dentures in terms of functionality and biocompatibility in all the aforementioned clinical situations. Our results have shown that all all-ceramic systems, especially densely sintered zirconia and reinforced glass ceramics, have a promising future to satisfy both practitioners and patients. However, technical and biological complications need to be taken into account when planning multi-unit fixed partial dentures for the posterior areas.

**Abstract:**

The metal-ceramic fixed partial prosthesis is the golden standard for posterior tooth restorations. Following the demands of patients and clinicians for metal-free restorations, all-ceramic materials were developed as they offer an adequate alternative with better optical qualities and good mechanical properties. This study aims to carry out a bibliographic review to assess the survival rate and the biological and technical complications of all-ceramic and metal-ceramic fixed partial dentures. An electronic search for articles in the English language literature was performed using PubMed (MEDLINE). This literature review focused on research studies between 2010 and 2020 that performed clinical studies on tooth-supported fixed partial dentures with a mean follow-up of at least 3 years. All the studies, which analyzed the survival and complications of tooth-supported fixed partial dentures, were included. Thus, 14 studies reporting on 756 all-ceramic and 160 metal-ceramic fixed partial dentures met the inclusion criteria. A comparative analysis was carried out based on all the data existing in the studies included in this review. The metal-ceramic fixed partial dentures showed survival rates of 95% to 100% at 3-, 5-, and 10-year follow-up periods. Zirconia fixed partial dentures were reported to have survival rates of 81% to 100% at 3-, 5-, 9-, 10-year follow-up evaluations. The reinforced glass-ceramic fixed partial dentures showed survival rates of 70% to 93.35% at 5 years, while the alumina FPDs showed a survival rate of 68% at 3 years follow-up. The incidence of caries and loss of vitality were reported as higher for all-ceramic prostheses as compared to the metal-ceramic ones. A significant framework fracture was reported for glass-infiltrated alumina fixed partial dentures in comparison to metal-ceramic fixed partial dentures. All-ceramic and metal-ceramic restorations showed similar survival rates after 3 years, although all-ceramic restorations have problems with technical complications such as chipping, which can lead to framework fractures over time.

## 1. Introduction

The treatment of edentulism with fixed partial dentures (FPDs) by replacing missing teeth is well established and provides promising clinical outcomes. The metal-ceramic FPDs are still perceived as the golden standard for posterior tooth restorations. They provide excellent mechanical properties but lack aesthetic characteristics due to the dark framework underneath, which has to be veneered and can be challenging in areas with insufficient space [1,2,3,4,5,6].

All-ceramic materials seem to offer an adequate alternative with better optical qualities, which are more tooth-resembling in terms of color and increased translucency [4,5,7,8,9,10]. In the past two decades, the science and technology of dental ceramics have witnessed rapid advances, representing the fastest-growing field of dental materials in research and development [11,12]. In this same time period, several types of ceramics and processing techniques were developed, and they enjoyed increased popularity with the end ceramic system and were given special credit through advances in CAD/CAM [6,13,14].

In order to satisfy the increased demand of patients and practitioners for high function, aesthetics, and biocompatibility regarding restorations, a broad spectrum of ceramics was introduced into medical practice [15]. Due to their mechanical properties, all-ceramics can be used as monolithic restorations (inlays, onlays, veneers, or crowns) [1,15,16,17,18]. Glass-ceramics and silicate ceramics are often used as veneering materials for metal-ceramic restorations or all-ceramic cores. For high load-bearing areas, the development of high-strength ceramics such as alumina and zirconia have helped establish qualitative good core materials for FPDs and crowns [1,13,15,19,20,21,22]. The growing interest in zirconia is based on the widening expansion of its applicability in medicine, especially in the field of prosthetic dentistry, in the last decades [23]. In its pure form, zirconia (ZrO_2_), which is an inorganic metallic oxide material, can be found in three temperature-dependent phases, namely: monolithic (up to 1170 °C), tetragonal (1170–2370 °C) or cubic (2370 °C up to the melting point) [24,25]. Because ZrO_2_ is unsuitable to be mechanically or structurally applied at room temperatures, adding yttrium-oxide (Y_2_O_3_) results in the stabilization of the dense tetragonal phase and, thus, presents yttria-stabilized zirconia (Y-TZP). The outcomes of the use of Y-TZP are promising due to its high flexure strength and fracture toughness in comparison with other ceramic core materials [26,27]. This explains why zirconia-based fixed dental prostheses, also well known as dental bridges, have a wider application than other ceramics; it is because they can be used in the posterior regions of the arch and in the molar area, respectively [23].

Irrespective of the materials from which they are made, the success rates of FPDs depend on the expected outcome and on how they perform within the stomatognathic system. Failure of restoration is defined as any circumstance leading to replacement [3,28,29,30]. Such conditions of failure may include biological factors (i.e., secondary decay, irreversible pulpitis), chemical factors (i.e., erosion and roughening of the ceramic surface) or mechanical factors (i.e., excessive wear of antagonistic surfaces, ditching of the cement margin, cracking, chipping, bulk fracturing or inadequate aesthetics) [3,31,32,33,34,35,36].

To conclude, considering the above circumstances, the aim of our study is to provide a literature review of the current evidence for the efficacy of metal-ceramics and all-ceramics in the treatment of edentulous spaces in the posterior areas of the dental arches. Moreover, we aim to compare the survival rates of metal-ceramic and all-ceramic bridges in terms of functionality and biocompatibility in all the aforementioned clinical situations.

## 2. Materials and Methods

An electronic search for articles in English language literature was performed using Pub Med (MEDLINE). This literature review focused on research articles between 2010 and 2020, while the following searches and search terms were applied: all-ceramic FPDs, zirconia, lithium disilicate, metal-ceramic FPDs, porcelain fused to metal, metal-ceramic restoration, all-ceramic restoration, all-ceramic and metal-ceramic, FPD esthetic, FPD biologic, FPD function (Figure 1).

The inclusion criteria for the study selection were: randomized controlled trials; prospective longitudinal studies and retrospective longitudinal studies reported posteriorly; FPDs containing details on the characteristics of FPDs, with a minimum follow-up period of 3 years. All full-text articles with studies of the type mentioned above and in which the survival and complications of metal-ceramic and all-ceramic tooth-supported FPDs were analyzed were also included in the current review. The exclusion criteria: papers about studies in vitro or animal studies related to implant-supported FPDs, only single-tooth restorations, and frontal restorations, as well as studies with less than 3 years follow-up.

The electronic search process was systematically conducted within two reviews in three stages. In the first stage, 9720 titles were identified based on the keywords. These titles were screened for meeting the inclusion criteria. Following this process, 9342 titles were excluded. In the second stage, the abstracts of the remaining titles were analyzed, and, consequently, 335 studies were excluded because they were considered inappropriate upon the review of the title and abstract or because they were duplications. In the third stage, the full-text articles were analyzed based on the inclusion/exclusion criteria. Thus, the material and methods, results, and discussions sections of these studies were thoroughly analyzed. The 29 out of 43 full-text articles were excluded, which brought the final number of this review to 14 articles, with the view of analyzing the survival and complications of metal-ceramic and all-ceramic tooth-supported posterior FPDs (Figure 2).

Further, a comparative analysis has been developed based on all data existing in the studies included in this review. The data was entered in spreadsheet software (Microsoft Excel, version 2013, Microsoft Office, Redmond, Washington, U.S.); the results were expressed numerically and as a percentage using this software. The following descriptive parameters were extracted: author(s), publication year, framework material, study design, observation period, number and mean age of patients. Number of FPDs, mean follow-up time, number of failures, estimated annual failure rate and estimated survival after N years, biological complications including secondary caries, periodontal disease, loss of vitality, abutment fracture, technical complications including framework or core fracture, chipping, ceramic fracture and loss of retention were also recorded for each of the 14 eligible studies.

The FPDs were evaluated as survived if they were present with/without complications during the whole observation period. Additionally, they were evaluated as successful if the FPD did not present any biological or technical complications at the time of the follow-up. Failure rates were calculated by dividing the number of events (failures) in the numerator by the total FDP exposure. The numerator could usually be extracted directly from the publication. The total exposure time was calculated by taking the sum of: exposure time of FDPs that could be followed for the whole observation time. For each study, event rates for the FDPs were calculated by dividing the total number of events by the total FDP exposure time in years.

## 3. Results

### 3.1. Characteristics of the Studies

Overall, 14 studies published between 2010 and 2020 met the inclusion criteria of the present literature review, respectively (Table 1); 3 studies were carried out on lithium disilicate glass-ceramic, 1 study on lithium disilicate glass-ceramic and zirconia, 5 studies on zirconia, 4 studies on zirconia and metal-ceramic, and 1 study on zirconia, metal-ceramic, and alumina (Table 1). The median value of the year in which the studies were published was 2016. Most of the studies were carried out in an academic environment—11 out of 14, to be precise. The remaining studies have been conducted in either specialist clinics or private practices. Additionally, the 14 eligible studies included a total number of 916 patients. Of these, 756 patients had been treated with all-ceramic FPDs and 160 patients with metal-ceramic FPDs. The age of the patients ranged between 16 and 87 years at the time when the treatments were performed (Table 1).

The majority of the studies (6) were prospective (42.9%) [39,40,42,44,48,50], followed by 5 randomized controlled clinical trials (35.7%) [37,38,45,46,47] and, lastly, 3 studies (21.4%) with a retrospective longitudinal design [41,43,49] (Figure 3).

The majority of the studies for all-ceramic FPDs had a prospective and retrospective design, and one was a randomized clinical control trial (RCT). The five randomized clinical control trials [37,38,45,46,47] compared various types of AC-FDPs (all-ceramic fixed partial dentures) with MC-FPDs (metal-ceramic fixed partial dentures). Most of these randomized controlled trials included zirconia and one lithium disilicate glass-ceramic as core material. Furthermore, the all-ceramic materials that were evaluated in the trials were made out of densely sintered zirconia (Y-TZP), glass infiltrated alumina-zirconia (InCeram Alumina), and reinforced glass-ceramics (lithium disilicate) (Table 1). The studies reported on the MC-FPDs included framework material out of high-noble (gold) and base metal (cobalt chromium).

Most of the studies evaluated densely sintered zirconia (49%), followed by metal-ceramic (25%), reinforced glass-ceramic (21%), and glass-infiltrated alumina (5%) (Figure 4).

The systems that were used and evaluated in the included studies were: Procera (Nobel Biocare, Gothenburg, Sweden), Cercon (Dentsply Friadent, Mannheim, Germany), Lava (Pre-Sintered) (3M ESPE, Seefeld, Germany), DC-Zircon (HIP) [41], CAM systems, manually fabricated frameworks making mock-ups and FPDs produced by milling Cercon brain (DeguDent, Hanau, Germany) sintered densely for 6 h at 1350 °C [42], presintered Cercon Zirconia, Everest, Lava (3M ESPE, Seefeld, Germany), IPS e.max ZirCAD, Wol-Ceram (Alumina) (Wol-Dent, Bad Soberheim, Germany), IPS e.max ZirPress (Ivoclar Vivadent Ellwangen, Germany) [37], Denzir HIPed Y-TZP (Detronic AB skellefteå) [43], Cercon-Ceram (Dentsply Friadent, Mannheim, Germany [45], abutments digitized with InEos scanner (Dentsply Sirona, York, PA, USA), VITA VM 13 (VITA Zahnfabrik, Bad Säckingen, Germany) [47], Lava Ultimate (3M, Seefeld, Germany) [48], and IPS e.max lithium disilicate CAD-CAM (Ivoclar Vivadent Ellwangen, Germany) [49]. Metal-ceramic restorations were fabricated in laboratories cooperating with the universities or private practices by means of the lost-wax technique or simple wax mock-ups with base-metal and high-noble framework materials.

### 3.2. Survival of Fixed Partial Dentures

For metal-ceramic FPDs, 5 out of 14 studies provided data on 160 FPDs after a mean follow-up period of 5.2 years. The number of total failures was 2. The mean survival rate after 3, 5, and 10 years ranged from 95% to 100%, with an estimated annual failure rate of 0.72–1.84%. Four studies reported survival rates of 100% even after an observation period of 10 years. The mean-survival rate was 98.28%, with an estimated annual mean failure rate of 0.46% (Table 2).

For all-ceramic FPDs, the results were divided into reinforced glass-ceramic (lithium disilicate), glass-infiltrated alumina (InCeram), and densely sintered zirconia (Y-TZP). For the reinforced glass ceramics, 4 studies offered data on 211 FPDs; 24 restorations had to be replaced due to failure. Some of these FPDs also got lost after a mean follow-up period of 7 years. The survival rate after five years ranged from 70% to 95.35%, with a mean survival rate of 86.66%. The annual failure rates ranged from 0.93% to 7.14%, and the median value of the annual failure rate was estimated to be 3.0 (Table 2).

One study [37] provided, alongside densely sintered zirconia and metal-ceramic FPDs, additional data on glass-infiltrated alumina. A total number of 34 glass-infiltrated alumina FPDs were examined in the study, and the given data were evaluated. After an observational period of 3 years, 11 FPDs were reported as failures. The estimated annual failure rate was 10.6%, with a survival rate of 68% after 3 years (Figure 4).

For densely sintered zirconia (Y-TZP), 10 studies contributed with 511 FPDs, from which 33 were reported as failures after an average follow-up period of 6.3 years. The survival rates ranged from 85% to 100% after observation periods from 3 to 10 years. The estimated annual failure rate had values ranging between 0% and 4.66%, with a mean value of 1.31%. The mean survival rate was 92.17% (Table 2).

The mean survival rates after 5 years were 100% for metal-ceramic FPDs, confirming the survival rates from the literature [37,49] and stating them as a reference. Restorations made from densely sintered zirconia performed similarly (92.2%) after a mean observation period of 5 years, and this confirmed comparable conclusions [46], namely: the fact that densely sintered zirconia can be used as an alternative to metal-ceramic FPDs. Reinforced glass-ceramics have also achieved comparable results, with a mean survival rate of 86.7% after 5 years. Glass-infiltrated alumina FPDs had a survival rate of 68% after 3 years, the lowest of the reported survival rates (Figure 5).

The annual failure rates varied from 0% to 10.6% and the survival rates from 68% to 100% for the types of FPDs (Table 2). Metal-ceramic FPDs have been used as a reference regarding their outcome, while all-ceramic FPDs showed lower survival rates and increased annual failure rates. Furthermore, the results for glass-infiltrated alumina have shown the most deficient values in terms of annual failure rates (Table 2). Another observation is that three of the studies have reported similar results regarding survival rates between metal-ceramic and all-ceramic FPDs, with slightly increased occurrences of chipping or fracture [45,46,47]. Out of the 14 studies, 2 studies reported inferior properties in all-ceramic restorations compared to metal-ceramic restorations due to increased incidences of chipping or fractures, resulting in decreased survival rates [37,50].

### 3.3. Biological Complications

#### 3.3.1. Secondary Caries

From a total number of 916 FPDs, secondary caries occurred on 26 abutments (Table 3). Densely sintered zirconia FPDs had most of the reported infestations of secondary caries on 20 abutments, whereas reinforced glass-ceramic FPDs showed only 3 carious abutments and metal-ceramic FPDs showed 3 carious abutments. The 4 studies [37,43,44,45] provided data about abutment fracture due to secondary caries with consequent FPD failure. Even though most of the affected abutments have been from densely sintered zirconia FPDs, the outcomes have shown to be a non-significant 2.8% of all FPDs that have failed due to secondary caries. As a result, there is no major difference in comparison to the FPD type of fixed partial denture in terms of secondary caries (Table 3).

From the evaluated studies, 6 of them reported cases of abutments losing their vitality throughout the treatment after cementation. Loss of abutment vitality occurred in 19 FPDs, with the majority (15 abutments) being all-ceramic FPDs, namely: densely sintered zirconia (Table 3).

#### 3.3.2. Periodontal Disease

Abutments of FPDs that were lost due to periodontal disease were reported in 9 FPDs. The prevalence was 6 all-ceramic FPDs and 3 metal-ceramic FPDs. Of the all-ceramic FPDs that were reported as failures, five were made from densely sintered zirconia and one FPD from reinforced glass-ceramic (Table 3).

#### 3.3.3. Abutment Tooth Fracture

The occurrence of failure due to abutment tooth fracture was reported in 9 abutments that were lost during the observational period. All of the reported failures were all-ceramic FPDs, whereas seven abutment teeth fractures were made of densely sintered zirconia and two were made of reinforced glass-ceramic (Table 3).

### 3.4. Technical Complications

#### 3.4.1. Framework or Core Fracture

The occurrence of framework or core fracture was reported in 8 out of the 14 studies evaluated in this systematic review. The number of reported failures due to framework fracture was 64 out of 916 fixed partial dentures (6.98%). The affected FPDs were all made from all-ceramic materials, including 16 FPDs made from densely sintered zirconia, 11 restorations made from reinforced glass-ceramic, and 11 made from alumina (Table 4).

Even though most of the lost FPDs were made from densely sintered zirconia, alumina frameworks were reported with a higher prevalence in framework fractures (Table 4), considering their low number, which we also referred to. Metal-ceramic FPDs showed no framework fractures in all the results.

#### 3.4.2. Chipping

The occurrence of chipping was reported in the majority of the studies included in this literature review; 194 FPDs showed chipping as a complication with different degrees of severity. The majority of chipped FPDs were all-ceramic restorations, with a total number of 164. Most of them were made from densely sintered zirconia, followed by alumina and reinforced glass-ceramic (Table 4). Compared to the number of FPDs, a percentage of 50% chipping frequency was reported in the case of alumina, while the lowest occurrence was recorded in the case of reinforced glass-ceramic. On the other hand, a number of 30 metal-ceramic FPDs with chipping as a complication was reported. However, most cases were considered to be minor chipping.

#### 3.4.3. Ceramic Fracture

The incidence of ceramic fracture occurred as another technical complication in 120 FPDs. Most of the fractures occurred in all-ceramic FPDs—in 51 densely sintered zirconia FPDs, 6 reinforced glass-ceramic FPDs, and one glass infiltrated alumina FPD, to be precise. Four metal-ceramic restorations were reported with ceramic fractures.

#### 3.4.4. Loss of Retention

Loss of retention or decementation was reported in 4 studies out of the studies included in the present review. Out of the total of 916 FPDs, 21 restorations needed recementation. The majority of these FPDs were made from densely sintered zirconia (Table 4). One study suggested reconsideration regarding the use of zinc-phosphate cements for the conventional luting of zirconia FPDs because of a high decementation rate [42]. Furthermore, resin-based cement was reported to show better results in combination with densely sintered zirconia and reinforced glass-ceramic, as shown by the results of the studies evaluated in the current review [10,42,44,46,47,49].

However, despite significant developments in adhesive protocol toward enamel and dentine, failures related to secondary caries are still a major issue when adhesive restorations are addressed. It should be considered that, prior to the appearance of secondary caries, interfacial gap formation plays an important role as it represents the first sign of restoration deterioration [51].

## 4. Discussion

The evaluated studies included in this literature review confirmed that all-ceramic restorations have similar survival rates after 3 years compared to the golden standard, the metal-ceramic FPD. The mean-survival rates after 5 years clearly show that densely sintered zirconia and reinforced glass-ceramic can compete with metal-ceramic FPDs [52]. The differences in the clinical outcomes between all-ceramic and metal-ceramic FPDs were observed after 5 years of function.

The reinforced glass-ceramic FPDs showed survival rates of 70% to 93% at 5 years. The metal-ceramic FPDs have shown to have 3-, 5-, and 10-year survival rates of 95% to 100%. Zirconia FPDs were shown to have survival rates ranging from 81% to 100% at 3-, 5-, 10-year follow-up evaluation, while the alumina FPDs had a survival rate of 68% at 3 years. This finding is in accordance with the findings of other studies that compared the survival rates of all-ceramic and metal-ceramic FPDs [53,54,55].

Rinke et al. [42] reported increased failure and complication rates of all-ceramic restorations between 3 and 7 years of their follow-ups. The majority of the failures were caused by technical complications: chipping, ceramic fracture, and framework fractures. However, the review of the studies included in the present review showed technical complications in both categories, namely: metal-ceramic and all-ceramic FPDs, which led to failure or replacement. Chipping was one of the primary complications of all-ceramic FPDs in this literature review. Even though minor chipping was reported almost equally in both types of restorations, major chipping occurred more often in zirconia or reinforced glass-ceramic FPDs [46,56]. Ioannidis et al. [44] stated that minor and major chippings of the veneering ceramic of the FPDs have a 4.9 higher probability in 4- and 5-unit FPDs than in 3-unit FPDs.

Regarding framework fractures, they occurred only in all-ceramic restorations, predominantly in alumina FPDs and densely zirconia FPDs in the present review. In this respect, Sulaiman et al. observed a low failure rate for lithium disilicate FPDs in a follow-up period of up to 7.5 years, which is in accordance with our results [57]. Koenig et al. [41] reported that the occurrence of framework fracture was always localized at the junction between the connector and the pontic FPD and depended on the following parameters: the number of elements, the nature of antagonists, and the type of support, which showed high significance apart from these complications. To sum up, most of the reported technical complications were chipping, with a few cases of replacement. Each study had different scales for ranging the severity of chipping. In all the studies that reported chipping, densely sintered zirconia FPDs were the most numerous. However, most of the cases could be polished or repaired if the degree of chipping was minor to medium. One study [41] that evaluated zirconia restorations proved through fractographic analysis that the origins of fractures were, in almost all cases, on the occlusal surfaces of the restorations; the study highlighted the importance of external stress, for example, the functional stress regarding the occurrence of chipping. Most of the veneer fractures originated from occlusal roughness, which was also confirmed by Koenig et al. [41], who observed that the stress was related to the patient parameter. This stress is added to the residual stress developed during the manufacturing process. As a result, their study highlighted the importance of mechanical behavior, which is barely influenced by manufacturing and material choice. The authors suggest that patients wear a night guard in order to minimize the occurrence of chipping or fractures.

An important observation of our review was that ceramic chipping was the most frequently reported complication, followed by ceramic fracture and core fracture. Chippings can be polished when they are detected early in order to prevent further worsening, which can lead to ceramic or framework fractures. The occurrence of technical complications is highly dependent on design, the all-ceramic system, as well as the cementation and present antagonists of similar materials. Loss of retention was much more prevalent in zirconia-based restorations, although the statistical outcome is not significant [42,46].

We take into account that most of the studies followed strict exclusion criteria when planning the trials and that not all parameters that could lead to technical complications related to the parafunctional habits were included. Patients who continued attending their follow-ups showed surprisingly good results in one of the studies, with even better results from the baseline examination [50]. The gathered data can be useful for future treatment plans when considering different treatment options and more materials.

From the evaluated studies, 3 trials on densely sintered zirconia reported survival rates of 100% after 3 and even 5 years. However, frequent check-ups and early detection of complications such as chipping, which were polished or repaired, are necessary in order to maintain a high survival rate [45,46,47]. Furthermore, each study used different classifications when reporting the incidence of chipping or fractures. All used a ranking system ranging between four values, i.e., Alpha to Delta or A to D, in which minor chips of <1 mm in diameter could be left alone or polished to the catastrophic loss of veneering [43]. One of these classifications, used by Håff et al. [43] and published by Crisp et al. [58], could be used in future trials to evaluate technical complications more accurately and would serve as a reference to other studies.

Biological complications such as secondary caries and loss of tooth vitality were more often seen in all-ceramic restorations, especially in zirconia. Similar results regarding the appearance of secondary caries have been reported by Pjetursson et al. [53]; thus, higher caries prevalence was reported on all-ceramic restorations in comparison to metal-ceramic FPDs. Loss of vitality was reported to be a possible cause due to an increased tooth reduction when preparing for an all-ceramic restoration. In this respect, Ioannidis et al. reported that the loss of vitality is the most frequent biological complication, with 7% rate of incidence, assuming that it could be due to the amount of tooth loss during crown preparation, which is 68% to 78% of lost tooth substance in posterior restorations [44]. Other authors [59,60,61] have shown a significant association between the loss of vitality and the cementation type. Both conventional and adhesive cementation could be used for all-ceramic restorations, but the latter is considered a highly sensitive technique and susceptible to contamination.

Furthermore, the adhesive cementation requires multiple steps to prepare the tooth surface for restoration. Good isolation and proper etchability of the enamel/dentin are necessary. Additionally, the quality of the bond is amendable to surface conditioning, which has been proved by applying different methods of cleaning to restoration after etching. The effects of the adhesive cementation on clinical outcomes may be quite different, such as loss of vitality and the marginal discrepancies that are represented by the internal gap. Prior to clinical failure, usually considered as the debonding restoration, the internal gap is the first sign; it can lead to bacterial recolonization of the tooth crown and root tooth system and, with subsequent endodontic failure [51].

Some of the authors [49,53] have suggested a conclusion that further investigations are required in order to determine more accurate survival rates. Aesthetic evaluations for posterior FPDs have been seldom evaluated on a broader scale, which has led to the results being more focused on biological and technical complications. Many studies focused on implant-based fixed partial dentures and single-tooth restorations, which are more frequent than the studies evaluating posterior all-ceramic restorations. This was one of the main reasons for the exclusion.

On the other hand, several studies evaluated the survival rate of all-ceramic restorations and came to results of 100% after 3 or even 5 years. However, most of these studies have excluded patients with parafunctional habits, resulting in higher survival rates [41]. This can be debatable because all-ceramic restorations are contraindicated in patients with parafunctional habits such as bruxism. Other authors [1,54,56], who have compared all-ceramic restorations to metal-ceramic restorations, reported a higher incidence of technical complications than biological failures. Pjetursson et al. [53] stated that densely sintered zirconia is more stable than reinforced glass-ceramic and alumina, but metal-ceramic fixed partial dentures were still reported as having the lowest failure rates. Moreover, Anusavice [28] suggested a unified classification system for describing chipping fractures more accurately in order to expand quantitative descriptions for future trials when monitoring ceramic prostheses and in order to reduce the frequency of chipping fractures.

Practitioners have to consider many factors such as cementation, occlusion, opposed teeth of all-ceramic material, parafunctional habits, and aesthetics when planning a treatment in order to satisfy the patient’s needs. In order to ensure a high survival rate, check-ups of the restorations are essential for detecting minor chippings in time in order to keep the longevity of the restoration. Despite the limitations of our study (the long span, the number of abutments involved, the occlusion, antagonists of FPDs), our data also suggest improving existing all-ceramic systems to reduce the incidence of technical or biological complications.

## 5. Conclusions

In this literature review, most of the studies have shown similar results regarding the survival and success rates of all-ceramic and metal-ceramic FPDs. The technical and biological complications also had similar outcomes. All-ceramic and metal-ceramic restorations show similar survival rates after 3 years, although all-ceramic restorations struggle with technical complications such as chipping, which can lead to framework fractures over time.

Zirconia and reinforced glass-ceramic systems proved to be able compete with the golden standard of metal-ceramic restorations. Thus, they also show promising results for the future.

Alumina restorations were reported to be less promising in the posterior areas due to the increased risk of the framework fracture. Despite technical complications, all-ceramic systems showed promising results even after 5 years. However, more studies are required to strengthen the hypothesis of all-ceramic systems being a definitive alternative to metal-ceramic posterior restorations.

To conclude, all-ceramic systems, especially densely sintered zirconia and reinforced glass-ceramics, have a promising future to satisfy both practitioners and patients. However, the technical and biological complications need to be taken into account when deciding to treat edentulism in the posterior areas.

## Figures and Tables

**Figure 1 biology-11-00556-f001:**
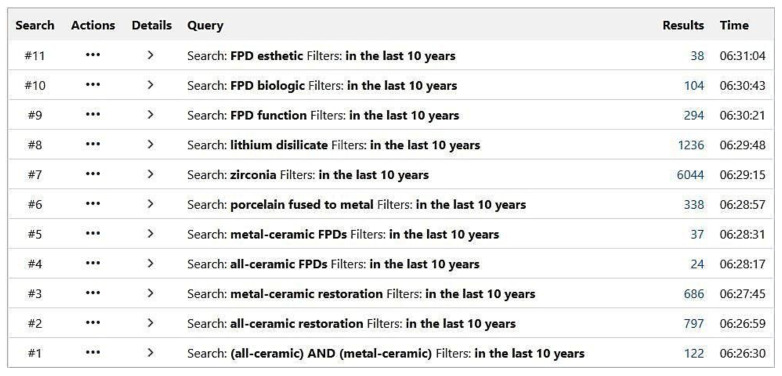
Search terms from Pubmed.

**Figure 2 biology-11-00556-f002:**
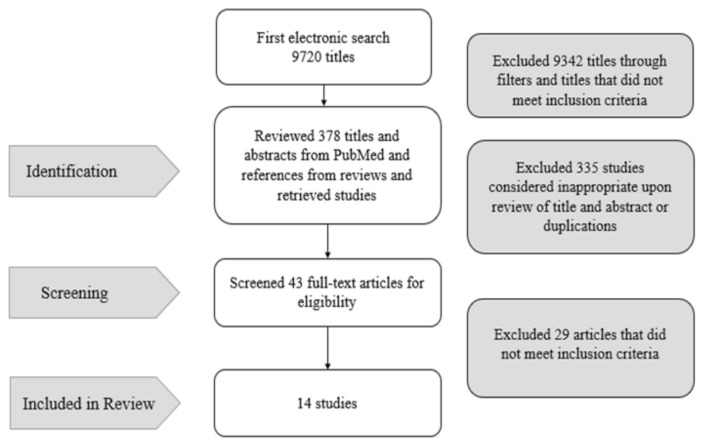
Flow chart of the search strategy.

**Figure 3 biology-11-00556-f003:**
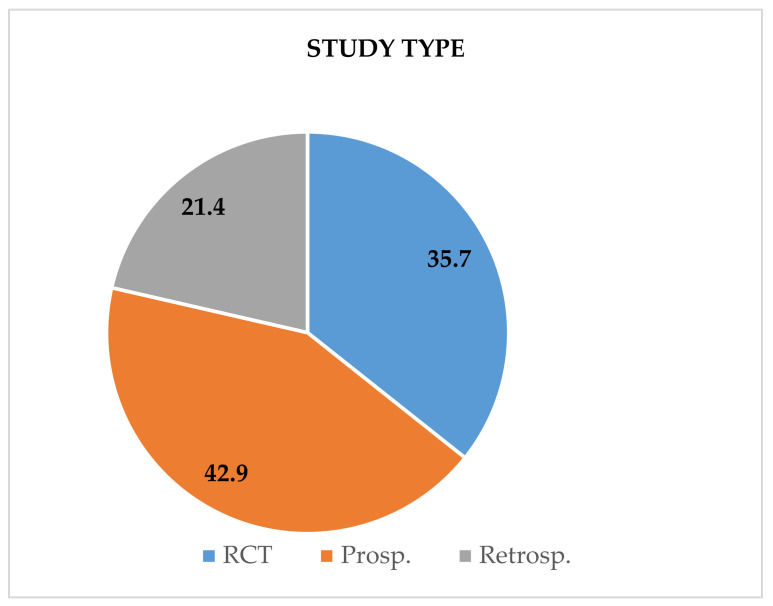
Types of the evaluated studies (RCT- Randomized control trial; Prosp- Prospective longitudinal study; Retrosp-Retrospective longitudinal study).

**Figure 4 biology-11-00556-f004:**
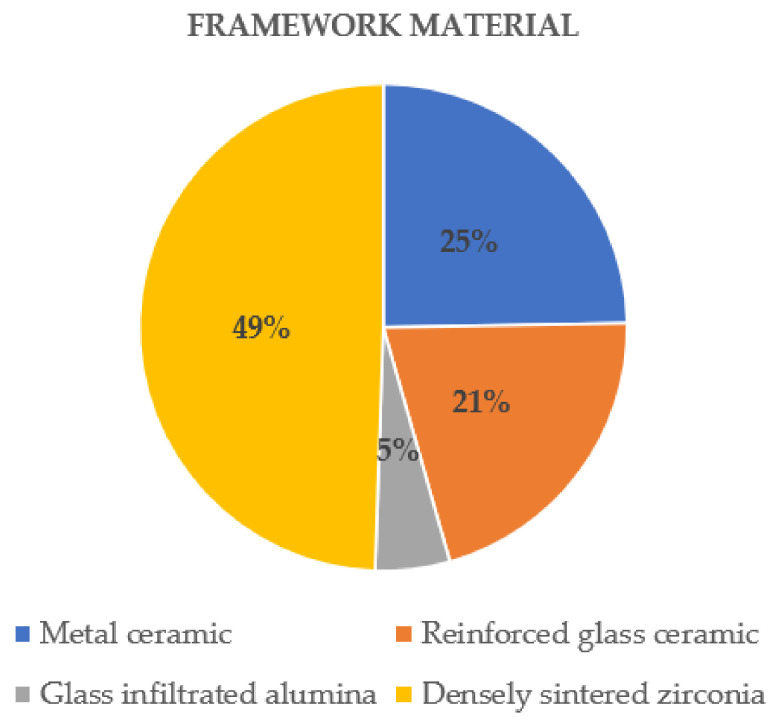
Distribution of the evaluated framework materials.

**Figure 5 biology-11-00556-f005:**
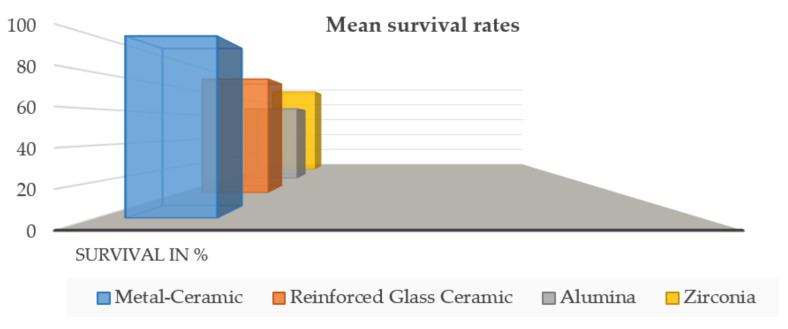
Mean survival rates after 5 years.

**Table 1 biology-11-00556-t001:** Characteristics of the reviewed studies (RCT- Randomized control trial; Prosp-Prospective longitudinal study; Retrosp- Retrospective longitudinal study, n.r.- not reported).

Study	Year	Framework Material	Study Design	Observation Period (Y)	No. of Patients in Study	Age Range	Mean Age	Setting
Christensen et al. [37]	2010	Zirconia, metal, and alumina	RCT	3	266	16–89	50	Specialists and private practice
Makarouna et al. [38]	2011	Lithium disilicate	RCT	6	18	n.r.	47	University
Kem et al. [39]	2012	Lithium disilicate	Prosp.	10	36	n.r.	47.5	University
Sola-Ruiz et al. [40]	2013	Lithium disilicate	Prosp.	10	21	n.r.	49	University
Koenig et al. [41]	2013	Zirconia	Retrosp.	9	30	25–79	54.6	University
Rinke et al. [42]	2013	Zirconia	Prosp.	7	99	26–76	49.4	University
Haff et al. [43]	2015	Zirconia	Retrosp.	13	33	35–87	68 ± 11	Specialists and private practice
Ioannidis et al. [44]	2016	Zirconia	Prosp.	10	57	n.r.	52.6 ± 10.1	University
Nicolaisen et al. [45]	2016	Zirconia, high-noble metal	RCT	3	34	36–66	51	University
Sailer et al. [46]	2018	Zirconia, high-noble metal	RCT	10	53	36.5–86.9	60.9	University
Suarez et al. [47]	2019	Zirconia, Co-Cr metal	RCT	5	40	24–70	n.r.	University
Koenig et al. [48]	2019	Zirconia	Prosp.	3	10	n.r.	54.34	University
Brandt et al. [49]	2019	Zirconia, lithium disilicate	Retrosp.	5	136	30–70	57.84	Specialists and private practice
Forrer et al. [50]	2020	Zirconia, high-noble metal	Prosp.	5	83	n.r.	n.r.	University

**Table 2 biology-11-00556-t002:** Survival of metal-ceramic and all-ceramic fixed partial dentures.

Study	Year	Total No. of FPDs	Mean Follow-Up Time	No. of Failures	Estimated Annual Failures	Estimated Survival in % after (N) = Years
**Metal-ceramic**						
Chirstensen et al. [37]	2010	69	n.r.	0	1.6	(3) 95%
Nicolaisen et al. [45]	2016	17	n.r.	0	0	(3) 100%
Sailer et al. [46]	2018	24	10.0	1	0	(10) 100%
Suarez et al. [47]	2019	20	5.25 ± 0.2	0	0	(5) 100%
Forrer et al. [50]	2020	30	6.44 ± 1.14	1	0.72	(5) 96.4%
**Reinforced glass-ceramic**						
Makarouna et al. [38]	2011	18	4.7	6	7.14	(5) 70%
Kem et al. [39]	2012	36	10.1	4	1.10	(5) 94.6%
Sola-Ruiz et al. [40]	2013	21	10.0	6	2.86	(5) 86.7%
Brandt et al. [49]	2019	136	3.10 ± 1.5	8	0.93	(5) 9.35%
**Glass-infiltrated alumina**						
Christensen et al. [37]	2010	34	n.r.	11	10.6	(3) 68%
**Zirconia**						
Christensen et al. [37]	2010	163	n.r.	2	4.66	(3) 86%
Koenig et al. [41]	2013	30	3.96 ± 2.7	1	2.04	(9) 81.6%
Rinke at al. [42]	2013	99	6.3	19	3.03	(5) 85.9%
Haff et al. [43]	2015	33	9.6 ± 1.6	2	0.6	(10) 94%
Ioannidis et al. [44]	2016	57	6.3 ± 1.9	3	1.5	(10) 85%
Nicolaisen et al. [45]	2016	17	n.r.	0	0	(3) 100%
Sailer et al. [46]	2018	29	10.3	5	0.87	(10) 91.3%
Saurez et al. [47]	2019	20	5.25 ± 0.2	0	0	(5) 100%
Koenig et al. [48]	2019	10	n.r.	1	0	(3) 100%
Forrer et al. [50]	2020	53	6.44 ± 1.14	2	0.42	(5) 97.9%

**Table 3 biology-11-00556-t003:** Biological complications of metal-ceramic and all-ceramic fixed partial dentures.

FPD Type	Total Number of FPDs	Total Number of Abutments Affected Due to:			
		a. Secondary Caries	b. Periodontal Disease	c. Loss of Vitality	d. Abutment Fracture
All FPDs	916	26	9	19	9
MC-FPDS	160	3	3	1	0
AC-FPDs	756	23	6	18	9
a. Densely sintered Zirconia	511	20	5	15	7
b. Reinforced Glass Ceramic	211	3	1	3	2
c. Alumina	34	0	0	0	0

**Table 4 biology-11-00556-t004:** Technical complications of metal-ceramic and all-ceramic fixed partial dentures.

Items	Total Number of FPDs	Framework or Core Fractures	Chipping	Ceramic Fractures	Loss of Retention
All FPDs	916	32	194	62	21
MC-FP Ds	160	0	30	4	3
AC-FPDs	756	32	164	58	19
a. Zirconia	511	16	141	51	15
b. Reinforced Glass Ceramic	211	5	6	6	4
c. Alumina	34	11	17	1	0

## Data Availability

The data presented in this study are available from the corresponding author upon reasonable request.

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
