# Peer review of "Comparative Assessment of the Functional Parameters for Metal-Ceramic and All-Ceramic Teeth Restorations in Prosthetic Dentistry—A Literature Review"

_biology, 2022, doi:10.3390/biology11040556_

Round 1

Reviewer 1 Report

The review is interesting and can be a contribution to the scientific field.

However, I suggest some changes in order to improve the overall quality of the manuscript for the readers.

This is not a systematic review, there is no risk assessment and other issues in the methodology.

So, as stated in the title, it is a literature review.

A single database is a limited search, a review shall at least include Scopus, Embase, Web of Science and Pubmed.

Line 20 in the abstract:

the authors make reference to a systematic review, but this is not a systematic review. This is also well stated in the title (literature review) please change accordingly

Line 52:

The authors make reference to 3 papers for CAD/CAM articles.

I would suggest to change reference 14 (which is related to optical properties) to another paper more specific to CAD/CAM restorations such for example as:

Baldi A, Comba A, Michelotto Tempesta R, Carossa M, Pereira GKR, Valandro LF, Paolone G, Vichi A, Goracci C, Scotti N. External Marginal Gap Variation and Residual Fracture Resistance of Composite and Lithium-Silicate CAD/CAM Overlays after Cyclic Fatigue over Endodontically-Treated Molars. Polymers (Basel). 2021 Sep 4;13(17):3002. doi: 10.3390/polym13173002. PMID: 34503042; PMCID: PMC8434150.

The authors should add in a table the search strategy used. “all-ceramic FPDs, zirconia FPDs, lithium disilicate FPDs, metal-ceramic FPDs, porcelain fused to metal, metal-ceramic restoration, all-ceramic restoration, all-ceramic and metal-ceramic. “ is too generic.

Line 83: same issue as line 20

line 91: same issue as line 20

line 104: same issue as line 20

line  192: same issue as line 20

Line 245-253: please outline the importance of the precision of the cemented restorations and the importance of a precise seal with the smallest gap possible. To support thi important aspect you could cite the following article: doi: https://doi.org/10.1111/jerd.12837

Line 311-313

The authors mentioned in the result the type of cement.

They should add a paragraph in the discussion describing the contribution of the type of cement to the failures of indirect restorations. (The authors just slightly mentioned issues related to cementation in lines 396-7)

Reviewer 2 Report

The manuscript submitted to Biology is a comprehensive review of the tha complication inherent to the FPDs presently in clinical use. It is an excellent overview of the state-of-the-art, and I am happy to recommend its publication in the journal.

Nevetheless, there are some minor modification that are mandatory before publication. 

1- line 182 : milling, NOT miling.

2 - line187: disilicate, NOT discilicate.

3 - line 217: EAFR = ?. Acronyms should be explained on their first occurrence....

4 - Figure 2: change in a pie graph (like figure 3). It will improve clarity

5 - Figure 4: enlarge histogram columns, insert space in between, .... !

6 - Revise format and check  names in Refs. # 1, 15, 18, 27.
